# What Decides the Well-Being of the Relationship between Parents and Adolescents

**DOI:** 10.3390/ijerph20010383

**Published:** 2022-12-26

**Authors:** Mimma Tafà, Fabrizia Bracaglia, Lucio Inguscio, Nicola Carone

**Affiliations:** 1Department of Dynamic, Clinical Psychology and Health Studies, Faculty of Medicine and Psychology, Sapienza University of Rome, 00185 Rome, Italy; 2Department of Brain and Behavioral Sciences, University of Pavia, 27100 Pavia, Italy

**Keywords:** family, families, new family forms, adolescent well-being, co-parenting, complexity, parent–adolescent relationship, protective and risk factors

## Abstract

The literature indicates a variety of factors that contribute to adolescent well-being: among these, the parent–adolescent relationship has a key role. The present article offered an overview of studies on parent–adolescent relationships across diverse family forms, not limited to the traditional family but also including “non-traditional” and “modern” families. To do so, this article described the evolution of the concept of family over the last fifty years and traced the significant family variables that guarantee adolescent well-being. Additionally, this article discussed the changes that occurred in family research, shifting from studies that considered only the family structure to more recent studies that investigated family processes and contextual factors. Overall, the reviewed studies indicated that the quality of parent–adolescent relationship, the interparental conflict and the consequent spillover effect on the child subsystem, and the changes in the economic situation following parental separation/divorce override the effect of the family structure. Finally, this article pointed out the need to examine, in the future research, adolescent well-being across diverse families by adopting more fine-grained methodologies, collecting data from the entire family system, and using a multi-method assessment to obtain a more ecological view of family complexity.

## 1. Introduction

The vast literature on how family dynamics affect child development mainly focused on the so-called “traditional family”, consisting of two cisgender, heterosexual, married, and fertile parents and their biological children, which was formerly considered the optimal environment for raising healthy children [1]. Such a traditional image of a middle-class family, with a bread-winning father and a stay-at-home mother, has certainly prevailed in past research comparing traditional and new family structures [1,2,3,4,5]. According to some authors [6], this comparative research line generally agreed that family structure influenced children’s well-being, following comparisons between traditional and separated families and emphasizing the advantages of the former over the latter [3,7,8,9]. In this vein, Amato [10] offered a review of the evolution of marriage in America since the second half of the last century, pointing out that children raised by two happily married people of different sexes have the best chance of becoming well-adjusted and successful adults [11,12,13].

Yet, current evidence indicates a significant increase in family forms that involve variation from this traditional pattern because the parents are diverse in their number, genders, gender identities, sexual orientations, and biological (un)relatedness with their child [14,15]. To date, little is known about the family characteristics that affect adolescent development and the quality of parent–adolescent relationships in diverse family forms, and whether these factors are those identified by research with the ideal, traditional families. The present article aims at filling the gap between research and prevailing opinion in the public debate by describing the limitations of previous research and offering an overview of the multiplicity of family forms, be they “non-traditional” (such as those formed through cohabitation or single parenthood, and step-parenthood following parental separation) or “modern” (such as sexual minority parent families, and heterosexual parent families formed through assisted reproduction) [14], that have been overlooked so far by research.

## 2. When the Traditional Family Held Primacy

Until the late 1960s, research focused largely on the traditional family, centered on two cisgender, heterosexual, procreative parents. This was considered the only functional family form. Apparently, the studies were looking for a factor that could explain and ensure the healthy growth of children. It followed that all family forms that deviated from the traditional one risked being considered pathological and, consequently, compromising children’s development: “Children are better off when their parents work to maintain the marriage. Consequently, society should make every effort to support healthy marriages and discourage married couples from divorcing” [9]. The following paragraphs examine some aspects of the family structure that have been the subject of disagreement.

### 2.1. Are Two Parents Better Than One?

In the traditional family, the positive effect of two parents on children’s well-being was strongly emphasized; without two parents, it was argued children’s psychosexual development and adjustment were at risk. Yet, research [1,16] has shown that the risks to children are due to negative factors that emerge during the family life cycle, such as financial difficulties and conflict between partners, and not to the family structure. For example, when controlling for a family’s income, family structure has, at most, a modest effect on children’s well-being [8,17].

### 2.2. From the Focus on the Family Structure to Attention to Family Processes

In the late 1980s, it became clear to many that children and adolescents fit well into both traditional and non-traditional families, such as stepparent families and single-parent families following divorce. Therefore, the focus on structural variables gradually gave way to attending to family process variables. The changes that characterized the family system and its relationships are well represented by Carter and McGoldrick’s Expanded Family Life Cycle [18], who revisited their classical life cycle theory [17] by extending it to new social emergencies (e.g., non-heterosexual orientation and multiculturalism) and thus to new family forms. With this in mind, the authors support the idea that it is not so much the family structure that determines adolescents’ well-being but the quality of affective relationships within it. For this reason, the present article overviewed several family structures to discuss family functioning and the relationship between parents and children.

In this vein, Kelly’s [19] highlighted the change in American research over the past three decades: on the one hand, he emphasized the general improvement in research methods, sample sizes, and advanced statistical analyses; on the other hand, he acknowledged that outcomes for children and adolescents after divorce were the product of multiple factors that must be considered simultaneously observed avoiding a deterministic reading. For the author, adverse factors, not considered by past research, were the real risk factors in a child’s development during separation. Demo and Acock [6] also found that family process variables (e.g., positive and negative mother–adolescent interaction) were better predictors of adolescent well-being than family structure. Similar results emerged from previous studies conducted in different socio-cultural contexts [20,21,22].

Several handbooks have been published on the topic of diverse contemporary families [23,24,25], and new variables for research have emerged, such as family functioning, quality of family life, parenting style, child–parent attachment, family cohesion, and co-parenting. The current scenario provides for a diverse family landscape, which means that a clinical assessment of the family can no longer be based solely on the structure, composition, and number of members: rather, it must consider the quality of ongoing relational processes (interactional, symbolic, and psychosocial), which have a significant influence on childhood development [20].

### 2.3. Separated Families and Their Evolution: Single-Parent and Reconstructed Families

The dissolution of the marital bond has become more frequent in most Western countries in the last 20 years. In 1980, Carter and McGoldrick published the family life cycle model [17] that described the image of the North American family as “normal” in those years. The model consists of six stages that the family encounters during its evolution (i.e., singlehood, couple, parenting, transformed by adolescence, middle life, and later life stages); these can be predictable, such as the birth of the first child and the subsequent adolescence, or unpredictable, such as a disabling accident of one of its members or a war.

As for Italy, Scabini [26] adapted the model by reducing the stages to five and suggested a fundamental question: should parental separation be considered a normative or paranormative event in the family cycle? Thus, a strong dialectic emerges. In fact, Scabini argued that the frequency of an event was not sufficient to make it normative; on the other hand, other authors stressed the impossibility of considering it an unpredictable event. Cigoli [27,28] overcame the impasse by calling it a historical event, i.e., part of the country’s culture and of which one cannot be unaware but which cannot be equated with an expected and predictable event even in part with the birth of a child or its adolescence, despite its strong statistical increase. Currently, marriages are broken off if they are not sustained by love; however, this is part of the physiology of a relationship, it is not a relational pathology or a “failure” according to the stereotype of the time, but a possible way to end a union. This is the case in Italy, but the increase in divorces has affected all European countries and even the United States: “marital commitment lasts only as long as people are happy and feel that their needs have been met” [10].

One aspect that has generally been in the research background was the application of a predominantly negative meaning to parental separation for child development [9,10,29]. In general, the research found that divorce damaged society and devastated children’s lives. In addition, social science research showed that the effects of divorce linger into adulthood and even hypothesized transmission to subsequent generations. In this regard, longitudinal research indicated that a child’s experience of divorce could affect their relational psychological well-being into adulthood [30]. In other words, the breakup of the parental couple increases the risk that children will divorce, thereby increasing the likelihood that they will exhibit behaviors that interfere with the maintenance of lasting and mutually rewarding intimate relationships [31]. Other authors [29] similarly argued that exposure to parental separation in childhood was associated with a “small but detectable increased risk of conduct, mood, and substance abuse disorders in adolescence”.

In the single-parent family, for example, research emphasized adolescents’ problems and vulnerabilities without considering their strengths [32]. This same research line has been driven by a deterministic comparison between a traditional family and a single-parent family, usually mother-led, whose separation was itself considered a negative outcome for the well-being of children and adolescents. Single-parent family structure has been associated with lower levels of parent–child interaction and parental supervision, support, and control [33,34]. Other authors [13] further emphasized the reduced chances of success for children born to a single parent, going so far as to state that “more than half of the children of the current generation will live in a single-parent family, and these children simply will not fare as well as their peers living with both parents”. However, these studies confused post-separation economic constraints with family structure. The exclusion of at least two factors certainly influenced the results. The economic one, for example, can lower a child’s standard of living in the transition from one to two homes. The other factor is the persistent conflict between the parents, which can make the children feel as if they are “dead” in the minds of the quarreling parents. In reality, conflict frequently represents the only way to defend oneself from the pain of separation [35]. This conflict is regularly linked to the child’s adjustment problems and indeed seems to be affected by them [36,37]. Of note, some authors recognized that children who experienced high levels of parental discord were not only at immediate psychological and physical risk but also at long-term developmental risk [38].

This high conflict can also produce what Katz and Gottman [39] called the “spillover effect”, namely the transfer of the positive or negative properties of one relationship to another. If the couple’s relationship is not positive and does not represent a source of satisfaction, it can create a hostile environment in which parents will be less likely to respond adequately to their children’s needs [40,41]. The spillover phenomenon confirms the association between marital status and parenting. McHale [42] extensively showed how the couple’s relationship influenced parenting and the construction of “co-parenting”. In addition, experts in family systems theory, which postulates reciprocity among family subsystems [42,43], identified the association between marriage and parenting in their family studies, which, in their view, appeared to be characterized by a constant bidirectional influence [44]. It should be clear, then, that it is the parental or ex-partner conflict, not the separation itself, that harms the child.

### 2.4. Co-Parenting

Parent–child relationships must be considered within a complex framework that includes the interdependence of the couple and their co-parenting relationship. To capture this complexity, McHale [42] referred to the term “co-parenting” or “co-parenting alliance” to refer to parents working together for their child. More precisely, co-parenting identifies the coordination between adults who support each other as family leaders and parents; it is not simply the sum of two independent parenting actions; rather, it is a two-way interdependent process in which the actions of one partner influence and influenced by those of the other [45]. Evidently, co-parenting is inversely proportional to conflict: the greater the conflict between partners, the less likely they are to be cooperative with each other [45]. In addition, co-parenting becomes a protective factor for children [46], especially when parental continuity is challenged and parents have separated. It is not entirely insignificant that intervention programs support healthy co-parenting and parenting practices aimed at parents who are going through separation and divorce [47]. Again, in relation to conflict, high levels of parental stress [48,49] and hostile couple conflict [50] are part of the overall emotional tone of separated parents and can also be transferred to the relationship with the child [51]. In particular, it is the mother’s parenting stress that most predicts an adolescent’s long-term aggressive behaviors [52].

Co-parenting, therefore, represents a key mechanism within the family system for predicting a child’s mental health after separation. Indeed, over the past decades, a great deal of evidence has accumulated to support the strong influence of interparental conflict on children’s emotional, physiological, and social behaviors [53,54,55]. Children exposed to high levels of interparental conflict are at high risk of developing health problems; however, their physical health has received much less attention. In this regard, a recent study [56] showed that co-parenting and positive parenting alleviated emotional problems and infectious diseases in children aged 3–8 years who have been exposed to parental conflict. In addition, a study [57] found that in separated families, the well-being of young children and adolescents was greater when parents maintained good co-parenting, sharing, talking to, and supporting each other. McHale emphasized the importance of co-parenting in different family structures. “Most knowledge about co-parenting is based on research conducted on heterosexual, married, European American or European two-parent families” [58] (p. 76). In doing so, he acknowledged that, even in different family structures, parenting is mediated not only by the mother and father but by all family members.

Similar considerations to those made so far can also be applied to another possible evolution of the separated family: the reconstituted family. This type of family has often been considered suboptimal, not only because of the result of separation but also because of the creation of non-biological but acquired parent–child bonds. This aspect is discussed later. The following paragraph discusses the family structure of the past and the controversies of traditionalists nostalgic for the traditional family.

### 2.5. Sexual Orientation of the Parents

In December 1973, homosexuality was removed as a mental disorder from the Diagnostic and Statistical Manual (DSM), first published in 1952. Up to that time, the motto “it can be cured” had influenced psychiatrists who advocated the admission of homosexual individuals to mental institutions. Heterosexual marriage was considered a truly effective cure for the “disease”, and the psychiatric hospital was the only avenue for those who rejected it. Despite the depathologization of homosexuality, in the 1990s, same-sex marriage was still not legally recognized worldwide, and families headed by lesbian or gay parents faced scorn and social intolerance. After the Netherlands extended the right to marry to same-sex couples in September 2000, the following decade saw a significant expansion of legal rights and recognition: same-sex marriage became legal in most countries worldwide, such as Argentina, Belgium, Brazil, Canada, France, Iceland, Mexico City, New Zealand, Norway, Portugal, Spain, South Africa, Sweden, the UK, Uruguay, and throughout the U.S.A. following the case Obergefell v. Hodges, in which the U.S. Supreme Court struck down all state bans on same-sex marriage, legalized it in all fifty states, and required states to honor out-of-state same-sex marriage licenses.

Yet, sexual minority parent families are still viewed as potentially problematic for healthy child development and against “the best interest of the child” (for a discussion, see [14]), particularly because the parents’ non-heterosexual orientation implies that children are raised by two same-gender parents. Such concerns have not been confirmed, however, by research conducted over the last 40 years in different socio-cultural contexts, indicating that children and adolescents raised by sexual minority parents are showing similar emotional, cognitive, social, and psychosexual development and academic success to those raised by heterosexual parents [14,59,60,61,62,63,64]. Based on these results, the leading associations of mental health professionals (e.g., American Academy of Pediatrics; American Psychological Association; American Psychiatry Association; British Psychological Society; Italian Association of Psychology) agreed that “despite economic and legal disparities and social stigma […] children’s well-being is affected much more by their relationships with their parents, their parents’ sense of competence and security, and the presence of social and economic support for the family than by the gender or the sexual orientation of their parents” [65] (p. e1374).

Research on the non-detrimental effects of parents’ sexual orientation on child development paralleled research on other structural family variables which make new family forms deviating from the traditional family, as extensively documented by Golombok’s research on non-biological parenthood in families formed through assisted reproduction or adoption, or on single parenthood by choice in families headed by single mothers through donor insemination [14,66]. Golombok [14] included these family forms among “modern families” to distinguish them from “non-traditional families” where the parents got separated or divorced. To summarize, this corpus of research not only challenged popular myths and assumptions about the presumed negative social and psychological consequences for children of growing up in new family structures but also consolidated traditional theories of child development, confirming that the quality of family relationships and the social environment are more influential for children’s psychological development than the number of parents, their gender, sexual orientation, or method of conception [16,63].

### 2.6. The Biological Bond between Parents and Children

The absence of a biological bond between parents and offspring was once thought to be a risk factor for children’s psychological problems; for example, parents in adoptive and reconstituted families were considered inherently “inadequate” simply because they were not the child’s biological parents. Therefore, research was conducted by comparing families created through gamete donation with those conceived spontaneously, in which, therefore, the child was biologically linked to both parents. Overall, results showed that children who did not have a biological link to their parents showed positive psychological adjustment and did not differ from those who shared a biological relationship [14,66].

Research with families created through assisted reproduction contributed to expanding what research with reconstituted families already had introduced, that is, a progressive distance and a not obvious correspondence between the parent–child biological link and the parental roles [67]. However, reconstituted families also share similar tasks with the biological family, though they are more complex. For example, the child will have four parents and grandparents, as well as possible half-siblings. Because of this, the dynamics of this family are closely related to those of the separated family from which it came. The way in which parents re-elaborate upon the experience of their separation is the factor that most influences a child’s outcomes in reconstituted families. As an example, a recent study [68] found that adult children’s bonds with both their biological father and their stepfather were interrelated and positively associated with the attitudes of the two fathers toward each other. Other longitudinal studies also reported that children’s views 20 years after separation were based on a constructive relationship between the parents [19].

With regard to adolescents, however, behavioral problems emerged in reconstituted families compared with nuclear families. The complexity [69] that distinguishes these families is their “polynuclearity”: the greater the number of nuclei, the greater the plurality of members. Therefore, family boundaries are highly variable: they can widen to include all nuclei but also shrink to enclose only part of the family (a biological parent and their children, or a biological parent, their partner, and the children of both). Finally, another feature that makes families complex is the coexistence of multiple parental figures [67]. It remains, however, that an adolescent’s sense of belonging seems to depend on the perceived quality of adolescent–stepparent relationships [70]. In addition, the flexibility and effectiveness of parent–adolescent communication are significantly related to parent–adolescent satisfaction [71].

### 2.7. Studies on Family Functioning

Research and clinical literature support the view that children’s well-being can be predicted by the quality of a family’s intergenerational relationships, regardless of the variability of its structure [6,20,72,73,74,75,76]. Recently, some authors [77] revisited the assumption of “family privilege” once enjoyed by the traditional family. In their longitudinal research, they appropriately used the word “stigma” to describe the experiences suffered by non-traditional and reaffirmed the importance of the quality of the parent–child relationship for the adjustment of children, regardless of family structure. Lamb, a leading author in developmental psychology, suggested that the clinical assessment of the family can no longer be based solely on its structure and composition but also, indeed, especially, on the quality of the relational processes that take place there [78]. In this vein, a more contemporary research focus has the merit of expanding the information obtainable by investigating family complexity and adolescent well-being [76].

More recently, several authors [73,74,79] were interested in examining positive family functioning, measured through an instrument designed by Olson [80]: the Family Adaptability and Cohesion Scale (FACES), now in its fourth version. Several research studies using the FACES instrument showed that adolescents’ well-being was related to family functioning rather than structure [79], as well as that Olson’s FACES-IV [81] core variables (i.e., family cohesion and flexibility) were associated with the quality of parent–adolescent relationship. In a study with adolescents belonging to diverse family structures (e.g., intact families, single-parent families, stepparent families), Lang et al. [79] showed that FACES-IV family cohesion predicted affective aspects of adolescent well-being, whereas cognitive elements of adolescent well-being were predicted by FACES-IV family flexibility, regardless of the family structure. In another study, a similar association was found between marital satisfaction, positive family relationships, and school performance in Chinese adolescents [82]. In addition, in a review of parental, familial, and contextual factors influencing children’s psychological adjustment, Lamb [1] compared numerous studies focused on studying children’s well-being in different families. The results indicated that children’s adjustment did not depend on the structural dimensions of the family (e.g., parental divorce, single parenthood, parental sexual orientation, parent–child biological ties) but rather on the overall quality of parenting, relationships between parents and their children, and available economic and social resources.

## 3. Conclusions

Past studies on the effect of family structure on adolescent development contributed to providing insights on which aspects related to family (i.e., structure, internal processes, contextual resources) mattered most for parents and their adolescents to flourish. Although this article did not aim at systematically reviewing studies on parent–adolescent relationship quality, it overviewed previous research to identify a complex series of contextual variables and family processes that explain the well-being of the adolescent, such as the interparental conflict [36,37,38,51,83], the consequent spillover effect on the child subsystem [40,41], the changes in the economic situation following parental separation/divorce [84], and the quality of new relationships in reconstituted families [68,70]. Additionally, this article examined evidence on some of the “modern families” identified by Golombok [14] (i.e., sexual minority parent families and families created through assisted reproduction), showing that neither the parents’ non-heterosexual orientation nor the parent–child biological unrelatedness has a detrimental effect on the adolescent development and the overall family relationships [15,59,63,66].

The attention to family process variables generated a fruitful shift of scientific interest in line with today’s reality, in which the nature and structure of families are changing significantly, prompted by more inclusive legislation, technological advancements, and shifts in societal attitudes toward family forms that were not possible, or even imagined, four decades ago. In this vein, aligning with previous non-systematic reviews (e.g., [1,66]), this article provided an updated discussion of the topic of interest, focusing on both studies with “non-traditional families” and studies with “modern families”. This is particularly relevant to advance the research field given that “non-traditional” elements may combine with “modern” elements, for example, when parents of children born through gamete donation divorce and remarry to form stepfamilies [14]. Whether such a combination will affect the adolescent well-being and the quality of parent–adolescent relationship remains, to date, largely unexplored. Additionally, given the shortage of studies on trans parent families, this article did not discuss the effect of parents’ gender identity on the adolescent–parent relationship. As the number of these families increases and children reach adolescence, future studies are warranted.

Another aspect highlighted by the present review is the importance of situating the parent–adolescent relationship within two key determinants of family well-being, namely co-parenting and family functioning. Therefore, this article emphasizes a meta-level embedded in the family, i.e., the parent–adolescent relationship, which represents a novel focus with respect to the study of the individual (i.e., the parent or the child), which has characterized past studies on family structure. For example, in the case of a separated family, the relationship level does not coincide with the behavior or the thoughts of the parent who leaves the other, nor with the child who suffers from the estrangement of a parent. Rather, the focus is the care of the relationship that is established or modified between the members of a family group.

In this vein, this review can offer a scientific basis from which to start for planning future studies that consider an evolution of the concept of the family but also the social changes of the last fifty years. Authors planning future research are advised to study relationships within the family, not just individual characteristics, as the well-being of the adolescent concerns several aspects: the parents, the child, and their relationship. It follows that in cases of intervention with families, the real patient is the relationship and not the individuals. Having in mind the different relationships that unfold within the family system, future studies should adopt more fine-grained and comprehensive methodologies of data collection (i.e., both parents and their adolescent, and potential siblings, where applicable; or nonparental caregivers who exert a parental role in the adolescents’ life) using a multi-method assessment (i.e., standardized questionnaires, semi-structured interviews, triadic and family observations). This approach would reflect the epistemology of complexity which uniquely reveals the effectiveness of relationships [85]. All in all, this review contributes to expanding information that can be obtained regarding the specifics of the different family structures. Given that the reviewed studies identified several family processes and contextual factors that significantly influence the parent–adolescent relationship across diverse family forms, future studies should examine family structure as a factor that can moderate, rather than predict, adolescent outcomes. 

To conclude, it is worth mentioning one of the key concepts offered by the anthropologist Bateson [86] to the systemic theory, namely the sense of connectedness or, in his words, the “pattern which connects the orchid to the primrose and the dolphin to the whale and all four to me”. Such a concept enables to synthesize all the multiple living forms because it is an aspect that can be traced in all of them and therefore unites them. Following this symbolic path, if we are looking for the structure that connects all the multiple current family forms, we could identify it in the *relation*, the pattern of Batesonian memory, which connects all aspects within and outside each peculiar family form, no matter how different they may seem. After all, again, with Bateson, it can be said that “only the difference and the comparison between differences is a matrix of knowledge” [86] to overcome the static nature of the traditional image of the family and point to deeply understanding the complexity of families.

## Data Availability

Not applicable.

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
