# Peer review of "What Decides the Well-Being of the Relationship between Parents and Adolescents"

_ijerph, 2022, doi:10.3390/ijerph20010383_

Round 1

Reviewer 1 Report

The article presented by the authors, however, presents methodological weaknesses in accordance with a review article and weaknesses in highlighting its importance as a work by omitting the discussion section.

1) It is necessary to address a systematization process of articles analyzed ("corpus"). This journal uses the PRISMA guidelines. (https://www.mdpi.com/journal/ijerph/instructions#preparation). After this standardized statement of your general article selection and the criteria for choosing documents, you can maintain the thematic and temporal order in which you have detailed your account. But this previous methodological phase is necessary.

2) You need to give an accounting of their review's contribution, how it "points out that the traditional family structure does not in itself guarantee the functional development of adolescents growing up within it". In this sense, the conclusion is not sufficient. They are required to detail a discussion with other review articles, explaining what is their contribution to pre-existing knowledge.

Author Response

To the reviewers

The additions and changes required are highlighted in red

The reviewer 1

The reviewer 1 speaks of 'methodological shortcomings' of the article: in this regard, at the end of the introduction it seemed appropriate to add the criteria -through wich- the articles used were as reviewer selected

The topic has indeed been dealt with, but the authors of this articles have not found a convincing systematization that this review would aspire to represent.

For what is missing, kindly contact dr Galina assistant editor.

Best regard

Reviewer 2 Report

The topic of the manuscript is interesting, a subject which is strongly researched in our days. As it is shown from the title and also from the abstract, the manuscript is focusing on presenting a review of studies on the topic of the relationship between parents and adolescents, meaning that it has a strong theoretical dimension. This could be very useful for other authors researching on the same subject, but could also limit the importance of the article. Our suggestion is that the authors might add a separate paragraph, at the end of the article (before Conclusions), in which they could present  some ideas of researches that could be developed in the future on this topic; or, some research hypotheses/ research questions they can formulate after this review of studies, and why not, they intend to develop in a new research (an empirical one, maybe) on the topic of the manuscript. 

Author Response

The topic of the manuscript is interesting, a subject which is strongly researched in our days. As it is shown from the title and also from the abstract, the manuscript is focusing on presenting a review of studies on the topic of the relationship between parents and adolescents, meaning that it has a strong theoretical dimension. This could be very useful for other authors researching on the same subject, but could also limit the importance of the article. Our suggestion is that the authors might add a separate paragraph, at the end of the article (before Conclusions), in which they could present some ideas of researches that could be developed in the future on this topic; or, some research hypotheses/ research questions they can formulate after this review of studies, and why not, they intend to develop in a new research (an empirical one, maybe) on the topic of the manuscript.

Many thanks for your appreciation. We followed your suggestion of including ideas for future research and capitalizing more the contribution of our article. However, to keep a consistent flow of the manuscript, we have integrated these points into a single Conclusion paragraph. We hope you agree. See as follows:

Past studies on the effect of family structure on adolescent development contributed to provide insights on which aspects related to family (i.e., structure, internal processes, contextual resources) mattered most for parents and their adolescents to flourish. Although this article did not aim at systematically reviewing studies on parent-adolescent relationship quality, it overviewed previous research to identify a complex series of contextual variables and family processes that explain the well-being of the adolescent, such as the interparental conflict [36,37,38,51,83], the consequent spillover effect on the child subsystem [40,41], the changes in the economic situation following parental separation/divorce [84], and the quality of new relationships in reconstituted families [68,70]. Also, this article examined evidence on some of the “modern families” identified by Golombok [14] (i.e., sexual minority parent families and families created through assisted reproduction), showing that neither the parents’ non-heterosexual orientation nor the parent-child biological unrelatedness have a detrimental effect on the adolescent development and the overall family relationships [15,59,63,66].

The attention to family process variables generated a fruitful shift of scientific interest in line with today’s reality, in which the nature and structure of families are changing significantly, prompted by more inclusive legislation, technological advancements, and shifts in societal attitudes towards family forms that were not possible, or even imagined, four decades ago. In this vein, aligning with previous non-systematic reviews [e.g., 1,66], this article provided an updated discussion of the topic of interest, focusing on both studies with “non-traditional families” and studies with “modern families”. This is particularly relevant to advance the research field given that “non-traditional” elements may combine with “modern” elements, for example, when parents of children born through gamete donation divorce and remarry to form stepfamilies [14]. Whether such combination will affect the adolescent well-being and the quality of parent-adolescent relationship remains, to date, largely unexplored. Also, given the shortage of studies on trans parent families, this article did not discuss the effect of parents’ gender identity on the adolescent-parent relationship. As the number of these families increase and children reach adolescence, future studies are warranted.

Another aspect highlighted by the present non-systematic review is the importance to situate the parent-adolescent relationship within two key determinants of family well-being, namely co-parenting and family functioning. So doing, this article emphasizes a meta-level embedded in the family, i.e. the parent-adolescent relationship, which represents a novel focus with respect to the study of the individual (i.e., the parent or the child), which has characterized past studies on family structure. For example, in the case of a separated family, the relationship level does not coincide with the behavior or the thoughts of the parent who leaves the other, nor with the child who suffers from the estrangement of a parent. Rather, the focus is the care of the relationship that is established or modified between the members of a family group.

In this vein, this non-systematic review can offer a scientific basis from which to start for planning future studies that consider an evolution of the concept of family, but also the social changes of the last fifty years. Authors planning future research are advised to study relationships within the family, not just individual characteristics, as the well-being of the adolescent concerns several aspects: the parents, the child, and their relationship. It follows that in cases of intervention with families, the real patient is the relationship and not the individuals. Having in mind the different relationships that unfold within the family system, future studies should adopt more fine-grained and comprehensive methodologies of data collection (i.e., both parents and their adolescent, and potential siblings, where applicable; or nonparental caregivers who exert a parental role in the adolescents’ live) using a multi-method assessment (i.e., standardized questionnaires, semi-structured interviews, triadic and family observations). This approach would reflect the epistemology of complexity which uniquely reveals the effectiveness of relationships [85].

All in all, this non-systematic review contributes to expand information that can be obtained regarding the specifics of the different family structures. Said it differently, given that the reviewed studies identified several family processes and contextual factors that significantly influence the parent-adolescent relationship across diverse family forms, future studies should examine family structure as a factor that can moderate, rather than predicts, adolescent outcomes. To conclude, it is worth mentioning one of the key concepts offered by the anthropologist Bateson [86] to the systemic theory, namely the sense of connectedness or, in his words, the “pattern which connects the orchid to the primrose and the dolphin to the whale and all four to me.” Such concept enables to synthesize all the multiple living forms, because it is an aspect that can be traced in all of them and therefore unites them. Following this symbolic path, if we are looking for the structure that connects all the multiple current family forms, we could identify it in the relation, the pattern of Batesonian memory which connects all aspects within and outside each peculiar family form, no matter how different they may seem. After all, again with Bateson, it can be said that “only the difference and the comparison between differences is a matrix of knowledge” [86] to overcome the static nature of the traditional image of the family and point to deeply understand the complexity of families.

Round 2

Reviewer 1 Report

Dear Authors, You MUST consider improvements to your article for it to be approved. The 2 improvements requested above require incorporating a methods section and a discussion section, which are strictly necessary for your article to be considered as a review article (I add again both requirements).

"... 1) It is necessary to address a systematization process of articles analyzed ("corpus"). This journal uses the PRISMA guidelines. (https://www.mdpi.com/journal/ijerph/instructions#preparation). After this standardized statement of your general article selection and the criteria for choosing documents, you can maintain the thematic and temporal order in which you have detailed your account. But this previous methodological phase is necessary.

2) You need to give an accounting of their review's contribution, how it "points out that the traditional family structure does not in itself guarantee the functional development of adolescents growing up within it". In this sense, the conclusion is not sufficient. They are required to detail a discussion with other review articles, explaining what is their contribution to pre-existing knowledge. ..."

Author Response

The article presented by the authors, however, presents methodological weaknesses in accordance with a review article and weaknesses in highlighting its importance as a work by omitting the discussion section.

  • It is necessary to address a systematization process of articles analyzed ("corpus"). This journal uses the PRISMA guidelines. (https://www.mdpi.com/journal/ijerph/instructions#preparation). After this standardized statement of your general article selection and the criteria for choosing documents, you can maintain the thematic and temporal order in which you have detailed your account. But this previous methodological phase is necessary.

Many thanks for your feedback. We believe there was a misunderstanding in the way we presented our article. Our aim was not to conduct a systematic review of the literature on parent-adolescent relationship. Rather, we aimed at critically discuss previous studies on this variable across diverse family forms. In this vein, we checked the journal guidelines and saw that for “review articles” (as the present article is), the PRISMA methodology is not required. To avoid any further confusion on this point, we have removed from the title “a review of studies”. Also, we have clearly stated in the Introduction and the Conclusion that the present article was not a systematic review. Finally, we noted that other non-systematic reviews are allowed by the International Journal of Environmental Research and Public Health, including a recent paper published within this Special Issue. We hope to have clarified this point.

  • You need to give an accounting of their review's contribution, how it "points out that the traditional family structure does not in itself guarantee the functional development of adolescents growing up within it". In this sense, the conclusion is not sufficient. They are required to detail a discussion with other review articles, explaining what is their contribution to pre-existing knowledge.

We recognize that this statement (i.e., "points out that the traditional family structure does not in itself guarantee the functional development of adolescents growing up within it") may sound causal. Therefore, we have extensively revised the abstract accordingly. Also, in the Conclusions section we have mentioned previous nonsystematic reviews and discussed the novel aspects and recommendation introduced by our article. See as follows:

[…] Although this article did not aim at systematically reviewing studies on parent-adolescent relationship quality, it overviewed previous research to identify a complex series of contextual variables and family processes that explain the well-being of the adolescent, such as the interparental conflict [36,37,38,51,83], the consequent spillover effect on the child subsystem [40,41], the changes in the economic situation following parental separation/divorce [84], and the quality of new relationships in reconstituted families [68,70]. Also, this article examined evidence on some of the “modern families” identified by Golombok [14] (i.e., sexual minority parent families and families created through assisted reproduction), showing that neither the parents’ non-heterosexual orientation nor the parent-child biological unrelatedness have a detrimental effect on the adolescent development and the overall family relationships [15,59,63,66].

[…] aligning with previous non-systematic reviews [e.g., 1,66], this article provided an updated discussion of the topic of interest, focusing on both studies with “non-traditional families” and studies with “modern families”. This is particularly relevant to advance the research field given that “non-traditional” elements may combine with “modern” elements, for example, when parents of children born through gamete donation divorce and remarry to form stepfamilies [14]. Whether such combination will affect the adolescent well-being and the quality of parent-adolescent relationship remains, to date, largely unexplored. Also, given the shortage of studies on trans parent families, this article did not discuss the effect of parents’ gender identity on the adolescent-parent relationship. As the number of these families increase and children reach adolescence, future studies are warranted.

Another aspect highlighted by the present non-systematic review is the importance to situate the parent-adolescent relationship within two key determinants of family well-being, namely co-parenting and family functioning. So doing, this article emphasizes a meta-level embedded in the family, i.e. the parent-adolescent relationship, which represents a novel focus with respect to the study of the individual (i.e., the parent or the child), which has characterized past studies on family structure. For example, in the case of a separated family, the relationship level does not coincide with the behavior or the thoughts of the parent who leaves the other, nor with the child who suffers from the estrangement of a parent. Rather, the focus is the care of the relationship that is established or modified between the members of a family group.

In this vein, this non-systematic review can offer a scientific basis from which to start for planning future studies that consider an evolution of the concept of family, but also the social changes of the last fifty years. Authors planning future research are advised to study relationships within the family, not just individual characteristics, as the well-being of the adolescent concerns several aspects: the parents, the child, and their relationship. It follows that in cases of intervention with families, the real patient is the relationship and not the individuals. Having in mind the different relationships that unfold within the family system, future studies should adopt more fine-grained and comprehensive methodologies of data collection (i.e., both parents and their adolescent, and potential siblings, where applicable; or nonparental caregivers who exert a parental role in the adolescents’ live) using a multi-method assessment (i.e., standardized questionnaires, semi-structured interviews, triadic and family observations). This approach would reflect the epistemology of complexity which uniquely reveals the effectiveness of relationships [85].

All in all, this non-systematic review contributes to expand information that can be obtained regarding the specifics of the different family structures. Said it differently, given that the reviewed studies identified several family processes and contextual factors that significantly influence the parent-adolescent relationship across diverse family forms, future studies should examine family structure as a factor that can moderate, rather than predicts, adolescent outcomes.

In light of the major edits we made to our article, please refer to the entire manuscript for getting the flow and the consistency of information contained in the article. Many thanks in advance.